# SimCKP: Simple Contrastive Learning of Keyphrase Representations

**Minseok Choi**[†*]    **Chaeheon Gwak**[‡]   **Seho Kim**[‡]   **Si Hyeong Kim**[‡]   **Jaegul Choo**[†]

[†]KAIST AI   [‡]Naver Webtoon AI

{minseok.choi,jchoo}@kaist.ac.kr
{ch.gwak,seho.kim,typekim}@webtoonscorp.com

## Abstract

Keyphrase generation (KG) aims to generate a set of summarizing words or phrases given a source document, while keyphrase extraction (KE) aims to identify them from the text. Because the search space is much smaller in KE, it is often combined with KG to predict keyphrases that may or may not exist in the corresponding document. However, current unified approaches adopt sequence labeling and maximization-based generation that primarily operate at a token level, falling short in observing and scoring keyphrases as a whole. In this work, we propose SIMCKP, a simple contrastive learning framework that consists of two stages: 1) An extractor-generator that extracts keyphrases by learning context-aware phrase-level representations in a contrastive manner while also generating keyphrases that do not appear in the document; 2) A reranker that adapts scores for each generated phrase by likewise aligning their representations with the corresponding document. Experimental results on multiple benchmark datasets demonstrate the effectiveness of our proposed approach, which outperforms the state-of-the-art models by a significant margin.[1]

## 1   Introduction

Keyphrase prediction (KP) is a task of identifying a set of relevant words or phrases that capture the main ideas or topics discussed in a given document. Prior studies have defined keyphrases that appear in the document as *present* keyphrases and the opposites as *absent* keyphrases. High-quality keyphrases are beneficial for various applications such as information retrieval (Kim et al., 2013), text summarization (Pasunuru and Bansal, 2018), and translation (Tang et al., 2016). KP methods are generally divided into keyphrase extraction (KE) (Witten

---

[*]Work done during an internship at Naver Webtoon.

[1]Our code is publicly available at https://github.com/brightjade/SimCKP.

**Title:** Nonlinear magnetostatic BEM formulation using one unknown double layer charge

**Document:** Purpose - The purpose of this paper is to solve generic magnetostatic problems by BEM, by studying how to use a boundary **integral equation** (BIE) with the double layer charge as unknown derived from the scalar potential. Design/methodology/approach - Since the double layer charge produces only the potential gap without disturbing the normal magnetic flux density, the field is accurately formulated even by one BIE with one unknown. Once the double layer charge is determined, Biot-Savart's law gives easily the magnetic flux density. Findings - The BIE using double layer charge is capable of treating robustly geometrical singularities at edges and corners. It is also capable of solving the problems with extremely high magnetic permeability. Originality/value - The proposed BIE contains only the double layer charge while the conventional equations derived from the scalar potential contain the single and double layer charges as unknowns. In the multiply connected problems, the excitation potential in the material is derived from the magnetomotive force to represent the circulating fields due to multiply connected exciting currents.

**Present keyphrases:** { boundary integral equation, double layer charge, multiply connected problem, scalar potential, integral equations }

**Absent keyphrases:** { nonlinear magnetostatic analysis, electric current }

**Sequence labeling result:** { boundary integral equation, double layer charge, multiply connected, scalar potential }

**MLE result:** { magnetostatic energy analysis, magnetomotive force, electric current }

Figure 1: An example of keyphrase prediction. Present and absent keyphrases are colored blue and red, respectively. Overlapping keyphrases are in **bold**.

et al., 1999; Hulth, 2003; Nguyen and Kan, 2007; Medelyan et al., 2009; Caragea et al., 2014; Zhang et al., 2016; Alzaidy et al., 2019) and keyphrase generation (KG) models (Meng et al., 2017; Ye and Wang, 2018; Chan et al., 2019; Chen et al., 2020b; Yuan et al., 2020; Ye et al., 2021b; Zhao et al., 2022), where the former only extracts present keyphrases from the text and the latter generates both present and absent keyphrases.

Recently, several methods integrating KE and KG have been proposed (Chen et al., 2019a; Liu et al., 2021; Ahmad et al., 2021; Wu et al., 2021, 2022b). These models predict present keyphrases using an extractor and absent keyphrases using a generator, thereby effectively exploiting a relatively small search space in extraction. However, current integrated models suffer from two limitations. First, they employ sequence labeling models that predict the probability of each token being a constituent of a present keyphrase, where such token-level predictions may be a problem when the target keyphrase is fairly long or overlapping. As shown in Figure 1, the sequence labeling model makes an incomplete prediction for the term "multi-

ply connected problem" because only the tokens for "multiply connected" have yielded a high probability. We also observe that the model is prone to miss the keyphrase "integral equations" every time because it overlaps with another keyphrase "boundary integral equation" in the text. Secondly, integrated or even purely generative models are usually based on maximum likelihood estimation (MLE), which predicts the probability of each token given the past seen tokens. This approach scores the most probable text sequence the highest, but as pointed out by Zhao et al. (2022), keyphrases from the maximum-probability sequence are not necessarily aligned with target keyphrases. In Figure 1, the MLE-based model predicts "magnetostatic energy analysis", which is semantically similar to but not aligned with the target keyphrase "nonlinear magnetostatic analysis". This may be a consequence of greedy search, which can be remedied by finding the target keyphrases across many beams during beam search, but it would also create a large number of noisy keyphrases being generated in the top-$k$ predictions.

Existing KE approaches based on representation learning may address the above limitations (Bennani-Smires et al., 2018; Sun et al., 2020; Liang et al., 2021; Zhang et al., 2022; Sun et al., 2021; Song et al., 2021, 2023). These methods first mine candidates that are likely to be keyphrases in the document and then rank them based on the relevance between the document and keyphrase embeddings, which have shown promising results. Nevertheless, these techniques only tackle present keyphrases from the text, which may mitigate the overlapping keyphrase problem from sequence labeling, but they are not suitable for handling MLE and the generated keyphrases.

In this work, we propose a two-stage contrastive learning framework that leverages context-aware phrase-level representations on both extraction and generation. First, we train an encoder-decoder network that extracts present keyphrases on top of the encoder and generates absent keyphrases through the decoder. The model learns to extract present keyphrases by maximizing the agreement between the document and present keyphrase representations. Specifically, we consider the document and its corresponding present keyphrases as positive pairs and the rest of the candidate phrases as negative pairs. Note that these negative candidate phrases are mined from the document using

a heuristic algorithm (see Section 4.1). The model pulls keyphrase embeddings to the document embedding and pushes away the rest of the candidates in a contrastive manner. Then during inference, top-$k$ keyphrases that are semantically close to the document are predicted. After the model has finished training, it generates candidates for absent keyphrases. These candidates are simply constructed by overgenerating with a large beam size for beam search decoding. To reduce the noise introduced by beam search, we train a reranker that allocates new scores for the generated phrases via another round of contrastive learning, where this time the agreement between the document and absent keyphrase representations is maximized. Overall, major contributions of our work can be summarized as follows:

- We present a contrastive learning framework that learns to extract and generate keyphrases by building context-aware phrase-level representations.

- We develop a reranker based on the semantic alignment with the document to improve the absent keyphrase prediction performance.

- To the best of our knowledge, we introduce contrastive learning to a unified keyphrase extraction and generation task for the first time and empirically show its effectiveness across multiple KP benchmarks.

## 2   Related Work

### 2.1   Keyphrase Extraction

Keyphrase extraction focuses on predicting salient phrases that are present in the source document. Existing approaches can be broadly divided into two-step extraction methods and sequence labeling models. Two-step methods first determine a set of candidate phrases from the text using different heuristic rules (Hulth, 2003; Medelyan et al., 2008; Liu et al., 2011; Wang et al., 2016). These candidate phrases are then sorted and ranked by either supervised algorithms (Witten et al., 1999; Hulth, 2003; Nguyen and Kan, 2007; Medelyan et al., 2009) or unsupervised learning (Mihalcea and Tarau, 2004; Wan and Xiao, 2008; Bougouin et al., 2013; Bennani-Smires et al., 2018). Another line of work is sequence labeling, where a model learns to predict the likelihood of each word being a keyphrase word (Zhang et al., 2016; Luan et al., 2017; Gollapalli et al., 2017; Alzaidy et al., 2019).

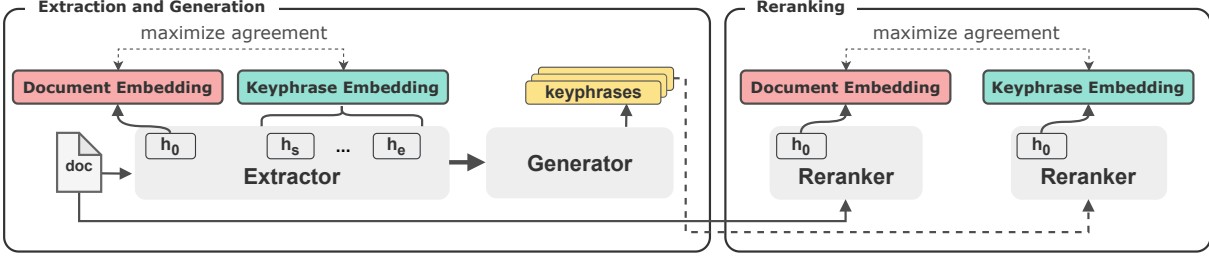

Figure 2: A contrastive framework for keyphrase prediction. In the first stage (left), the model learns to maximize the relevance between present keyphrases and their corresponding document while generating absent keyphrases. After training, the model generates candidates for absent keyphrases and sends them to the second stage (right), where the candidates are reranked after their relevance with the document has been maximized/minimized.

## 2.2 Keyphrase Generation

The task of keyphrase generation is introduced to predict both present and absent keyphrases. Meng et al. (2017) first proposed CopyRNN, a seq2seq framework with attention and copy mechanism under the ONE2ONE paradigm, where a model is trained to generate a single keyphrase per document. However, due to the problem of having to fix the number of predictions, Yuan et al. (2020) proposed the ONE2SEQ paradigm where a model learns to predict a dynamic number of keyphrases by concatenating them into a single sequence. Several ONE2SEQ-based models have been proposed using semi-supervised learning (Ye and Wang, 2018), reinforcement learning (Chan et al., 2019), adversarial training (Swaminathan et al., 2020), hierarchical decoding (Chen et al., 2020b), graphs (Ye et al., 2021a), dropout (Ray Chowdhury et al., 2022), and pretraining (Kulkarni et al., 2022; Wu et al., 2022a) to improve keyphrase generation.

Furthermore, there have been several attempts to unify KE and KG tasks into a single learning framework. These methods not only focus on generating only the absent keyphrases but also perform presumably an easier task by extracting present keyphrases from the document, instead of having to generate them from a myriad of vocabularies. Current methodologies utilize external source (Chen et al., 2019a), selection guidance (Zhao et al., 2021), salient sentence detection (Ahmad et al., 2021), relation network (Wu et al., 2021), and prompt-based learning (Wu et al., 2022b).

## 2.3 Contrastive Learning

Methods to extract rich feature representations based on contrastive learning (Chopra et al., 2005; Hadsell et al., 2006) have been widely studied in numerous literature. The primary goal of the learning process is to pull semantically similar data to be close while pushing dissimilar data to be far away in the representation space. Contrastive learning has shown great success for various computer vision tasks, especially in self-supervised training (Chen et al., 2020a), whereas Gao et al. (2021) have devised a contrastive framework to learn universal sentence embeddings for natural language processing. Furthermore, Liu and Liu (2021) formulated a seq2seq framework employing contrastive learning for abstractive summarization. Similarly, a contrastive framework for autoregressive language modeling (Su et al., 2022) and open-ended text generation (Krishna et al., 2022) have been presented.

There have been endeavors to incorporate contrastive learning in the context of keyphrase extraction. These methods generally utilized the pairwise ranking loss to rank phrases with respect to the document to extract present keyphrases (Sun et al., 2021; Song et al., 2021, 2023). In this paper, we devise a contrastive learning framework for keyphrase embeddings on both extraction and generation to improve the keyphrase prediction performance.

## 3 Problem Definition

Given a document $\mathbf{x}$, the task of keyphrase prediction is to identify a set of keyphrases $\mathcal{Y} = \{\mathbf{y}^i\}_{i=1,...,|\mathcal{Y}|}$, where $|\mathcal{Y}|$ is the number of keyphrases. In the ONE2ONE training paradigm, each sample pair $(\mathbf{x}, \mathcal{Y})$ is split into multiple pairs $\{(\mathbf{x}, \mathbf{y}^i)\}_{i=1,...,|\mathcal{Y}|}$ to train the model to generate one keyphrase per document. In ONE2SEQ, each sample pair is processed as $(\mathbf{x}, f(\mathcal{Y}))$, where $f(\mathcal{Y})$ is a concatenated sequence of keyphrases. In this work, we train for extraction and generation simultaneously; therefore, we decompose $\mathcal{Y}$ into a present keyphrase set $\mathcal{Y}_p = \{\mathbf{y}_p^i\}_{i=1,...,|\mathcal{Y}_p|}$ and an absent keyphrase set $\mathcal{Y}_a = \{\mathbf{y}_a^i\}_{i=1,...,|\mathcal{Y}_a|}$.

# 4 SIMCKP

In this section, we elaborate on our approach to building a contrastive framework for keyphrase prediction. In Section 4.1, we delineate our heuristic algorithm for constructing a set of candidates for present keyphrase extraction; in Section 4.2, we describe the multi-task learning process for extracting and generating keyphrases; and lastly, we explain our method for reranking the generated keyphrases in Section 4.3. Figure 2 illustrates the overall architecture of our framework.

## 4.1 Hard Negative Phrase Mining

To obtain the candidates for present keyphrases, we employ a similar heuristic approach from existing extractive methods (Hulth, 2003; Mihalcea and Tarau, 2004; Wan and Xiao, 2008; Bennani-Smires et al., 2018). A notable difference between prior work and ours is that we keep not only noun phrases but also verb, adjective, and adverb phrases, as well as phrases containing prepositions and conjunctions. We observe that keyphrases are actually made up of diverse parts of speech, and extracting only the noun phrases could lead to missing a significant number of keyphrases. Following the common practice, we assign part-of-speech (POS) tags to each word using the `Stanford POSTagger`[2] and chunk the phrase structure tree into valid phrases using the `NLTK RegexpParser`[3].

As shown in Algorithm 1, each document is converted to a phrase structure tree where each word $w$ is tagged with a POS tag $t$. The tagged document is then split into possible phrase chunks based on our predefined regular expression rules, which must include one or more valid tags such as nouns, verbs, adjectives, etc. Nevertheless, such valid tag sequences are sometimes nongrammatical, which cannot be a proper phrase and thus may introduce noise during training. In response, we filter out such nongrammatical phrases by first categorizing tags as *independent* or *dependent*. Phrases generally do not start or end with a preposition or conjunction; therefore, preposition and conjunction tags belong to a dependent tag set $\mathcal{T}_{dep}$. On the other hand, noun, verb, adjective, and adverb tags can stand alone by themselves, making them belong to an independent tag set $\mathcal{T}_{indep}$. There are also

---

[2]https://stanfordnlp.github.io/CoreNLP/
[3]https://www.nltk.org/

---

**Algorithm 1** Hard Negative Phrase Mining

**Input:** Source document $\mathbf{x}$, maximum n-gram length $n$, regular expression pattern $p$, POS tagging function $tag(\cdot)$, phrase parsing function $parse(\cdot)$, stemming function $stem(\cdot)$
**Output:** Present keyphrase candidate set $\mathcal{C}_{pre}$

1: $\mathcal{C}_{pre} \leftarrow \varnothing$
2: word_tag_pairs $\leftarrow tag(\mathbf{x})$
3: phrase_tree $\leftarrow parse(p, \text{word\_tag\_pairs})$
4: **for** phrase $\in$ phrase_tree **do**
5:     **for** $w_i, t_i \in$ phrase **do**
6:         **if** $t_i \notin \mathcal{T}_{indep}$ and $t_i \notin \mathcal{T}_{end\_dep}$ **then**
7:             **continue**
8:         $\text{span}_i \leftarrow stem(w_i)$
9:         $\mathcal{C}_{pre} \leftarrow \mathcal{C}_{pre} \cup \text{span}_i$
10:         **for** $w_{ij}, t_{ij} \in$ phrase$[i + 1:]$ **do**
11:             **if** len($\text{span}_i$.split()) $\geq n$ **then**
12:                 **break**
13:             **if** $t_{ij} \in \mathcal{T}_{dep}$ or $t_{ij} \in \mathcal{T}_{end\_dep}$ **then**
14:                 $\text{span}_i \mathrel{+}= stem(w_{ij})$
15:                 **continue**
16:             **if** $t_{ij} \in \mathcal{T}_{indep}$ or $t_{ij} \in \mathcal{T}_{start\_dep}$ **then**
17:                 $\text{span}_i \mathrel{+}= stem(w_{ij})$
18:                 $\mathcal{C}_{pre} \leftarrow \mathcal{C}_{pre} \cup \text{span}_i$
19: **return** $\mathcal{C}_{pre}$

---

tag sets $\mathcal{T}_{start\_dep}$ and $\mathcal{T}_{end\_dep}$, which include tags that cannot start but end a phrase and tags that can start but not end a phrase, respectively. Lastly, each candidate phrase is iterated over to acquire all n-grams that make up the phrase. For example, if the phrase is "applications of machine learning", we select n-grams "applications", "machine", "learning", "applications of machine", "machine learning", and "applications of machine learning" as candidates. Note that phrases such as "applications of", "of", "of machine", and "of machine learning" are not chosen as candidates because they are not proper phrases. As noted by Gillick et al. (2019), hard negatives are important for learning a high-quality encoder, and we claim that our mining accomplishes this objective.

## 4.2 Extractor-Generator

In order to jointly train for extraction and generation, we adopt a pretrained encoder-decoder network. Given a document $\mathbf{x}$, it is tokenized and fed as input to the encoder where we take the last hidden states of the encoder to obtain the contextual embeddings of a document:

$$[\mathbf{h}_0, \mathbf{h}_1, ..., \mathbf{h}_T] = \text{Encoder}_p(\mathbf{x}), \qquad (1)$$

where $T$ is the token sequence length of the document and $\mathbf{h}_0$ is the start token (e.g., ``) representation used as the corresponding document embedding. For each candidate phrase, we construct its embedding by taking the sum pooling of the token span representations: $\mathbf{h}_p =$

`SumPooling([h_s, ..., h_e])`, where $s$ and $e$ denote the start and end indices of the span. The document and candidate phrase embeddings are then passed through a linear layer followed by non-linear activation to obtain the hidden representations:

$$\mathbf{z}_{d_p} = \tanh(\mathbf{W}_{d_p}\mathbf{h}_0 + \mathbf{b}_{d_p})$$
$$\mathbf{z}_p = \tanh(\mathbf{W}_p\mathbf{h}_p + \mathbf{b}_p), \quad (2)$$

where $\mathbf{W}_{d_p}, \mathbf{W}_p, \mathbf{b}_{d_p}, \mathbf{b}_p$ are learnable parameters.

**Contrastive Learning for Extraction**  To extract relevant keyphrases given a document, we train our model to learn representations by pulling keyphrase embeddings to the corresponding document while pushing away the rest of the candidate phrase embeddings in the latent space. Specifically, we follow the contrastive framework in Chen et al. (2020a) and take a cross-entropy objective between the document and each candidate phrase embedding. We set keyphrases and their corresponding document as positive pairs, while the rest of the phrases and the document are set as negative pairs. The training objective for a positive pair $(\mathbf{z}_{d_p}, \mathbf{z}_{p,i}^+)$ (i.e., document and present keyphrase $y_p^i$) with $N_p$ candidate pairs is defined as

$$\mathcal{L}_{\text{CL}}^i = -\log \frac{e^{\text{sim}(\mathbf{z}_{d_p}, \mathbf{z}_{p,i}^+)/\tau}}{e^{\text{sim}(\mathbf{z}_{d_p}, \mathbf{z}_{p,i}^+)/\tau} + \sum_{j=1}^{N_p} e^{\text{sim}(\mathbf{z}_{d_p}, \mathbf{z}_{p,j}^-)/\tau}}, \quad (3)$$

where $\tau$ is a temperature hyperparameter and $\text{sim}(\mathbf{u}, \mathbf{v})$ is the cosine similarity between vectors $\mathbf{u}$ and $\mathbf{v}$. The final loss is then computed across all positive pairs for the corresponding document (i.e., $\mathcal{L}_{\text{CL}} = \sum_{i=1}^{|\mathcal{Y}_p|} \mathcal{L}_{\text{CL}}^i$).

**Joint Learning**  Our model generates keyphrases by learning a probability distribution $p_\theta(\mathbf{y}_a)$ over an absent keyphrase text sequence $\mathbf{y}_a = \{y_{a,1}, ..., y_{a,|\mathbf{y}|}\}$ (i.e., in an ONE2SEQ fashion), where $\theta$ denotes the model parameters. Then, the MLE objective used to train the model to generate absent keyphrases is defined as

$$\mathcal{L}_{\text{MLE}} = -\frac{1}{|\mathbf{y}_a|} \sum_{t=1}^{|\mathbf{y}_a|} \log p_\theta(y_{a,t}|\mathbf{y}_{a,<t}). \quad (4)$$

Lastly, we combine the contrastive loss with the negative log-likelihood loss to train the model to both extract and generate keyphrases:

$$\mathcal{L} = \mathcal{L}_{\text{MLE}} + \lambda\mathcal{L}_{\text{CL}}, \quad (5)$$

where $\lambda$ is a hyperparameter balancing the losses in the objective.

| Split | Dataset | #KP$_\mu$ | #KP$_\sigma$ | \|KP\|$_\mu$ | % Absent | # Samples |
|-------|---------|-----------|--------------|--------------|----------|-----------|
| Train | KP20k | 5.28 | 3.76 | 1.94 | 38.19 | 530,809 |
| Valid | KP20k | 5.26 | 3.67 | 1.94 | 38.26 | 20,000 |
| Test | KP20k | 5.27 | 3.74 | 2.04 | 37.03 | 20,000 |
| | Inspec | 9.81 | 4.97 | 2.33 | 22.25 | 500 |
| | Krapivin | 5.84 | 3.55 | 2.08 | 44.77 | 400 |
| | NUS | 10.85 | 6.67 | 2.13 | 51.55 | 211 |
| | SemEval | 14.97 | 3.50 | 2.15 | 55.74 | 100 |

Table 1: Dataset statistics. **#KP$_\mu$**: average number of keyphrases; **#KP$_\sigma$**: standard deviation of the number of keyphrases; **|KP|$_\mu$**: average length (n-gram) of keyphrases per document.

## 4.3  Reranker

As stated by Zhao et al. (2022), MLE-driven models predict candidates with the highest probability, disregarding the possibility that target keyphrases may appear in suboptimal candidates. This problem can be resolved by setting a large beam size for beam search; however, this approach would also result in a substantial increase in the generation of noisy keyphrases among the top-$k$ predictions. Inspired by Liu and Liu (2021), we aim to reduce this noise by assigning new scores to the generated keyphrases.

**Candidate Generation**  We employ the fine-tuned model from Section 4.2 to generate candidate phrases that are highly likely to be absent keyphrases for the corresponding document. We perform beam search decoding using a large beam size on each training document, resulting in the overgeneration of absent keyphrase candidates. The model generates in an ONE2SEQ fashion where the outputs are sequences of phrases, which means that many duplicate phrases are present across the beams. We remove the duplicates and arrange the phrases such that each unique phrase is independently fed to the encoder. We realize that the generator sometimes fails to produce even a single target keyphrase, in which we filter out such documents for the second-stage training.

**Dual Encoder**  We adopt two pretrained encoder-only networks and obtain the contextual embeddings of a document, as well as each candidate phrase $\mathbf{c}$: $[\mathbf{h}_d^0, \mathbf{h}_d^1, ..., \mathbf{h}_d^T] = \text{Encoder}_{a_1}(\mathbf{x})$ and $[\mathbf{h}_c^0, \mathbf{h}_c^1, ..., \mathbf{h}_c^{T_c}] = \text{Encoder}_{a_2}(\mathbf{c})$, where $T_c$ is the token sequence length of the candidate phrase and $\mathbf{h}_d^0$ and $\mathbf{h}_c^0$ are the start token representations used as the document and candidate phrase embedding, respectively. Consequently, their hidden representations are obtained by $\mathbf{z}_{d_a} = \tanh(\mathbf{W}_{d_a}\mathbf{h}_d^0 + \mathbf{b}_{d_a})$

| | Inspec | | Krapivin | | NUS | | SemEval | | KP20k | |
|---|---|---|---|---|---|---|---|---|---|---|
| Model | $F_1$@5 | $F_1$@M | $F_1$@5 | $F_1$@M | $F_1$@5 | $F_1$@M | $F_1$@5 | $F_1$@M | $F_1$@5 | $F_1$@M |
| *Generative Models* | | | | | | | | | | |
| catSeq (Yuan et al., 2020) | 0.225 | 0.262 | 0.269 | 0.354 | 0.323 | 0.397 | 0.242 | 0.283 | 0.291 | 0.367 |
| catSeqTG (Chen et al., 2019b) | 0.229 | 0.270 | 0.282 | 0.366 | 0.325 | 0.393 | 0.246 | 0.290 | 0.292 | 0.366 |
| catSeqTG-2$RF_1$ (Chan et al., 2019) | 0.253 | 0.301 | 0.300 | 0.369 | 0.375 | 0.433 | 0.287 | 0.329 | 0.321 | 0.386 |
| ExHiRD-h (Chen et al., 2020b) | 0.253 | 0.291 | 0.286 | 0.347 | – | – | 0.284 | 0.335 | 0.311 | 0.374 |
| SetTrans (Ye et al., 2021b) | 0.285 | 0.324 | 0.326 | 0.364 | 0.406 | 0.450 | 0.331 | 0.357 | 0.358 | 0.392 |
| CorrKG (Zhao et al., 2022) | 0.330 | **0.365** | – | – | 0.405 | 0.449 | 0.333 | 0.359 | 0.370 | 0.404 |
| *Unified Models* | | | | | | | | | | |
| SEG-Net (Ahmad et al., 2021) | 0.216 | 0.265 | 0.276 | 0.366 | 0.396 | 0.461 | 0.283 | 0.332 | 0.311 | 0.379 |
| UniKeyphrase (Wu et al., 2021) | 0.260 | 0.288 | – | – | 0.415 | 0.443 | 0.302 | 0.322 | 0.347 | 0.352 |
| PromptKP (Wu et al., 2022b) | 0.260 | 0.294 | – | – | 0.412 | 0.439 | 0.329 | 0.356 | 0.351 | 0.355 |
| SIMCKP | **0.356**$_6$ | 0.358$_8$ | **0.405**$_8$ | **0.405**$_8$ | **0.496**$_8$ | **0.498**$_9$ | **0.387**$_2$ | **0.386**$_4$ | **0.426**$_1$ | **0.427**$_1$ |

Table 2: Present keyphrase prediction results. The best results are in bold, while the second best are underlined. The subscript denotes the corresponding standard deviation (e.g., $0.427_1$ indicates $0.427 \pm 0.001$).

and $\mathbf{z}_a = \tanh(\mathbf{W}_a \mathbf{h}_c^0 + \mathbf{b}_a)$, where $\mathbf{W}_{d_a}$, $\mathbf{W}_a$, $\mathbf{b}_{d_a}$, $\mathbf{b}_a$ are learnable parameters.

**Contrastive Learning for Generation** To rank relevant keyphrases high given a document, we train the dual-encoder framework via contrastive learning. Following a similar process as before, we train our model to learn absent keyphrase representations by semantically aligning them with the corresponding document. Specifically, we set the correctly generated keyphrases and their corresponding document as positive pairs, whereas the rest of the generated candidates and the document become negative pairs. The training objective for a positive pair $(\mathbf{z}_{d_a}, \mathbf{z}_{a,i}^+)$ (i.e., document and absent keyphrase $y_a^i$) with $N_a$ candidate pairs then follows Equation 3, where the cross-entropy objective maximizes the similarity of positive pairs and minimizes the rest. The final loss is computed across all positive pairs for the corresponding document with a summation.

## 5 Experimental Setup

### 5.1 Datasets

We evaluate our framework on five scientific article datasets: **Inspec** (Hulth, 2003), **Krapivin** (Krapivin et al., 2009), **NUS** (Nguyen and Kan, 2007), **SemEval** (Kim et al., 2010), and **KP20k** (Meng et al., 2017). Following previous work (Meng et al., 2017; Chan et al., 2019; Yuan et al., 2020), we concatenate the title and abstract of each sample as a source document and use the training set of KP20k to train all the models. Data statistics are shown in Table 1.

### 5.2 Baselines

We compare our framework with two kinds of KP models: *Generative* and *Unified*.

**Generative Models** Generative models predict both present and absent keyphrases through generation. Most models follow **catSeq** (Yuan et al., 2020), a seq2seq framework under the ONE2SEQ paradigm. We report the performance of catSeq along with its variants such as **catSeqTG** (Chen et al., 2019b), **catseqTG-2RF₁** (Chan et al., 2019), and **ExHiRD-h** (Chen et al., 2020b). We also compare with two state-of-the-arts **SetTrans** (Ye et al., 2021b) and **CorrKG** (Zhao et al., 2022).

**Unified Models** Unified models combine extractive and generative methods to predict keyphrases. We compare with the latest models including **SEG-Net** (Ahmad et al., 2021), **UniKeyphrase** (Wu et al., 2021), and **PromptKP** (Wu et al., 2022b).

### 5.3 Evaluation Metrics

Following Chan et al. (2019), all models are evaluated on macro-averaged $F_1$@5 and $F_1$@$M$. $F_1$@$M$ compares all the predicted keyphrases with the ground truth, taking the number of predictions into account. $F_1$@5 measures only the top five predictions, but if the model predicts less than five keyphrases, we randomly append incorrect keyphrases until it obtains five. The motivation is to avoid $F_1$@5 and $F_1$@$M$ reaching similar results when the number of predictions is less than five. We stem all phrases using the Porter Stemmer and remove all duplicates after stemming.

### 5.4 Implementation Details

Our framework is built on PyTorch and Huggingface's Transformers library (Wolf et al., 2020). We use BART (Lewis et al., 2020) for the encoder-decoder model and uncased BERT (Devlin et al., 2019) for the reranking model. We optimize their weights with AdamW (Loshchilov and Hutter,

| Model | Inspec | | Krapivin | | NUS | | SemEval | | KP20k | |
|---|---|---|---|---|---|---|---|---|---|---|
| | $F_1$@5 | $F_1$@M | $F_1$@5 | $F_1$@M | $F_1$@5 | $F_1$@M | $F_1$@5 | $F_1$@M | $F_1$@5 | $F_1$@M |
| *Generative Models* | | | | | | | | | | |
| catSeq (Yuan et al., 2020) | 0.004 | 0.008 | 0.018 | 0.036 | 0.016 | 0.028 | 0.016 | 0.028 | 0.015 | 0.032 |
| catSeqTG (Chen et al., 2019b) | 0.005 | 0.011 | 0.018 | 0.034 | 0.011 | 0.018 | 0.011 | 0.018 | 0.015 | 0.032 |
| catSeqTG-2$RF_1$ (Chan et al., 2019) | 0.012 | 0.021 | 0.030 | 0.053 | 0.019 | 0.031 | 0.021 | 0.030 | 0.027 | 0.050 |
| ExHiRD-h (Chen et al., 2020b) | 0.011 | 0.022 | 0.022 | 0.043 | – | – | 0.017 | 0.025 | 0.016 | 0.032 |
| SetTrans (Ye et al., 2021b) | 0.021 | 0.034 | 0.047 | 0.073 | 0.042 | 0.060 | 0.026 | 0.034 | 0.036 | 0.058 |
| CorrKG (Zhao et al., 2022) | 0.032 | **0.045** | – | – | 0.061 | 0.079 | 0.039 | 0.044 | 0.053 | 0.071 |
| *Unified Models* | | | | | | | | | | |
| SEG-Net (Ahmad et al., 2021) | 0.009 | 0.015 | 0.018 | 0.036 | 0.021 | 0.036 | 0.021 | 0.030 | 0.018 | 0.036 |
| UniKeyphrase (Wu et al., 2021) | 0.026 | 0.036 | – | – | 0.045 | 0.056 | **0.045** | **0.052** | 0.046 | 0.068 |
| PromptKP (Wu et al., 2022b) | 0.017 | 0.022 | – | – | 0.036 | 0.042 | 0.028 | 0.032 | 0.032 | 0.042 |
| SIMCKP | **0.033**$_2$ | 0.035$_3$ | **0.078**$_1$ | **0.089**$_0$ | **0.076**$_{12}$ | **0.088**$_{15}$ | 0.040$_2$ | 0.047$_6$ | **0.073**$_2$ | **0.080**$_1$ |

Table 3: Absent keyphrase prediction results.

2019) and tune our hyperparameters to maximize $F_1$@$M$[4] on the validation set, incorporating techniques such as early stopping and linear warmup followed by linear decay to 0. We set the maximum n-gram length of candidate phrases to 6 during mining and fix $\lambda$ to 0.3 for scaling the contrastive loss. When generating candidates for absent keyphrases, we use beam search with the beam size 50. During inference, we take the candidate phrases as predictions in which the cosine similarity with the corresponding document is higher than the threshold found in the validation set. The threshold is calculated by taking the average of the $F_1$@$M$-maximizing thresholds for each document. If the number of predictions is less than five, we retrieve the top similar phrases until we obtain five. We conduct our experiments with three different random seeds and report the averaged results.

## 6 Results and Analyses

### 6.1 Present and Absent Keyphrase Prediction

The present and absent keyphrase prediction results are demonstrated in Table 2 and Table 3, respectively. The performance of our model mostly exceeds that of previous state-of-the-art methods by a large margin, showing that our method is effective in predicting both present and absent keyphrases. Particularly, there is a notable improvement in the $F_1$@5 performance, indicating the effectiveness of our approach in retrieving the top-$k$ predictions. On the other hand, we observe that $F_1$@$M$ values are not much different from $F_1$@5, and we believe this is due to the critical limitation of a global threshold. The number of keyphrases varies significantly for each document, and finding op-

---

[4]We compared with $F_1$@5 and found no difference in determining the best hyperparameter configuration.

| Method | In-domain | | Out-of-domain | |
|---|---|---|---|---|
| | $F_1$@5 | $F_1$@M | $F_1$@5 | $F_1$@M |
| *Present keyphrase prediction* | | | | |
| SIMCKP | **0.426** | **0.427** | **0.411** | **0.412** |
| w/o CL | 0.295 | 0.388 | 0.278 | 0.356 |
| CL$\Rightarrow$SEQLABEL | 0.236 | 0.384 | 0.180 | 0.272 |
| CL$\Rightarrow$BINARYCLF | 0.209 | 0.322 | 0.171 | 0.251 |
| *Absent keyphrase prediction* | | | | |
| SIMCKP | **0.073** | **0.080** | **0.057** | **0.064** |
| w/o CL | 0.030 | 0.066 | 0.027 | 0.051 |
| w/o RERANKER | 0.035 | 0.069 | 0.025 | 0.041 |

Table 4: Ablation study. "w/o CL" is the vanilla BART model using beam search for predictions. "w/o reranker" extracts with CL but generates using only beam search.

timal thresholds seems necessary for improving the $F_1$@$M$ performance. Nonetheless, real-world applications are often focused on identifying the top-$k$ keywords, which we believe our model effectively accomplishes.

### 6.2 Ablation Study

We investigate each component of our model to understand their effects on the overall performance and report the effectiveness of each building block in Table 4. Following Xie et al. (2022), we report on two kinds of test sets: 1) KP20k, which we refer to as **in-domain**, and 2) the combination of Inspec, Krapivin, NUS, and SemEval, which is **out-of-domain**.

**Effect of CL** We notice a significant drop in both present and absent keyphrase prediction performance after decoupling contrastive learning (CL). For a fair comparison, we set the beam size to 50, but our model still outperforms the purely generative model, demonstrating the effectiveness of CL. We also compare our model with two extractive methods: sequence labeling and binary classifica-

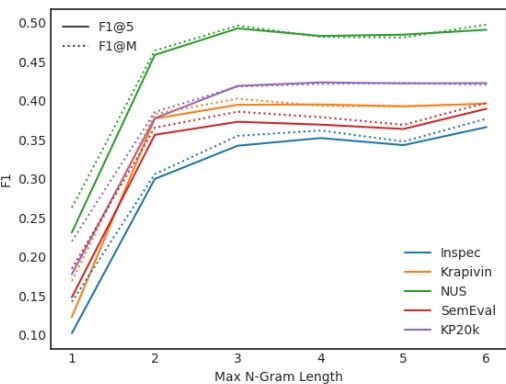

Figure 3: Comparison of present keyphrase prediction performance w.r.t max n-gram length during extraction.

| Mining Method | In-domain | | Out-of-domain | |
|---|---|---|---|---|
| | $F_1$@5 | $F_1$@M | $F_1$@5 | $F_1$@M |
| IN-BATCH DOC NEGATIVES | 0.077 | 0.038 | 0.092 | 0.065 |
| RANDOM NEGATIVES | 0.210 | 0.226 | 0.211 | 0.267 |
| HARD NEGATIVES (OURS) | **0.426** | **0.427** | **0.411** | **0.412** |

Table 5: Comparison of negative mining methods for present keyphrase prediction.

tion. For sequence labeling, we follow previous work (Tokala et al., 2020; Liu et al., 2021) and employ a BiLSTM-CRF, a strong sequence labeling baseline, on top of the encoder to predict a BIO[5] tag for each token, while for binary classification, a model takes each phrase embedding to predict whether each phrase is a keyphrase or not. CL outperforms both approaches, showing that learning phrase representations is more efficacious.

**Effect of Reranking** We remove the reranker and observe the degradation of performance in absent keyphrase prediction. Note that the vanilla BART (i.e., w/o CL) is trained to generate both present and absent keyphrases, while the other model (i.e., w/o RERANKER) is trained to generate only the absent keyphrases. The former performs slightly better in out-of-domain scenarios, as it is trained to generate diverse keyphrases, while the latter excels in in-domain since absent keyphrases resemble those encountered during training. Nevertheless, the reranker outperforms the two, indicating that it plays a vital role in the KG part of our method.

### 6.3 Performance over Max N-Gram Length

We conduct experiments on various maximum lengths of n-grams for extraction and compare the

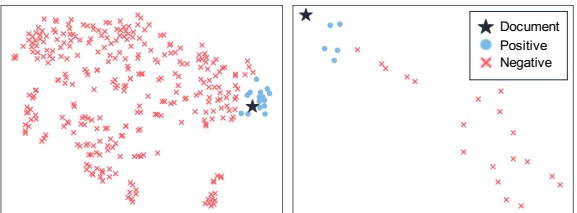

Figure 4: Visualization of the semantic space using t-SNE. The left shows the extractor space, while the right depicts the generator space after reranking.

present keyphrase prediction performance from unigrams to 6-grams, as shown in Figure 3. For all datasets, the performance steadily increases until the length of 3, which then plateaus to the rest of the lengths. This indicates that the testing datasets are mostly composed of unigrams, bigrams, and trigrams. The performance increases slightly with the length of 6 for some datasets, such as Inspec and SemEval, suggesting that there is a non-negligible number of 6-gram keyphrases. Therefore, the length of 6 seems feasible for maximum performance in all experiments.

### 6.4 Impact of Hard Negative Phrase Mining

In order to assess the effectiveness of our hard negative phrase mining method, we compare it with other negative mining methods and report the results in Table 5. First, utilizing in-batch document embeddings as negatives yields the poorest performance. This is likely due to ineffective differentiation between keyphrases and other phrase embeddings. Additionally, we experiment with using random text spans as negatives and observe that although it aids in representation learning to some degree, the performance improvement is limited. The outcomes of these two baselines demonstrate that our approach successfully mines hard negatives, enabling our encoder to acquire high-quality representations of keyphrases.

### 6.5 Visualization of Semantic Space

To verify that our model works as intended, we visualize the representation space of our model with t-SNE (van der Maaten and Hinton, 2008) plots, as depicted in Figure 4. From the visualizations, we find that our model successfully pulls keyphrase embeddings close to their corresponding document in both extractor and generator space. Note that the generator space displays a lesser number of phrases than the beam size 50 because the duplicates after stemming have been removed.

---

[5]We use the BIO format for our sequence labeling baseline. For example, if the phrase "voip conferencing system" is tokenized into "v ##oi ##p con ##fer ##encing system", it is labeled as "B I I I I I I".

| | Inspec | Krapivin | NUS | SemEval | KP20k |
|---|---|---|---|---|---|
| $R@50$ | $0.144_{16}$ | $0.226_5$ | $0.185_{12}$ | $0.072_5$ | $0.220_7$ |

Table 6: Upper bound performance for absent keyphrase prediction after overgeneration with the beam size 50.

## 6.6 Upper Bound Performance

Following previous work (Meng et al., 2021; Ray Chowdhury et al., 2022), we measure the upper bound performance after overgeneration by calculating the recall score of the generated phrases and report the results in Table 6. The high recall demonstrates the potential for reranking to increase precision, and we observe that there is room for improvement by better reranking, opening up an opportunity for future research.

## 7 Conclusion

This paper presents a contrastive framework that aims to improve the keyphrase prediction performance by learning phrase-level representations, rectifying the shortcomings of existing unified models that score and predict keyphrases at a token level. To effectively identify keyphrases, we divide our framework into two stages: a joint model for extracting and generating keyphrases and a reranking model that scores the generated outputs based on the semantic relation with the corresponding document. We empirically show that our method significantly improves the performance of both present and absent keyphrase prediction against existing state-of-the-art models.

## Limitations

Despite the promising prediction performance of the framework proposed in this paper, there is still room for improvement. A fixed global threshold has limited the potential performance of the framework, especially when evaluating $F_1@M$. We expect that adaptively selecting a threshold value via an auxiliary module for each data sample might overcome such a challenge. Moreover, the result of the second stage highly depends on the performance of the first stage model, directing the next step of research towards an end-to-end framework.

## Acknowledgements

This work was supported by Institute for Information & communications Technology Planning & Evaluation (IITP) grant funded by the Korea government (MSIT) (No. 2020-0-00368, A Neural-Symbolic Model for Knowledge Acquisition and Inference Techniques and No. 2021-0-02068, Artificial Intelligence Innovation Hub), and the National Research Foundation of Korea (NRF) grant funded by the Korea government (MSIT) (No. NRF-2022R1A2B5B02001913). We thank all researchers at NAVER WEBTOON Ltd. for their valuable discussions.

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

# A  Additional Details for SIMCKP

## A.1  Details for Hard Negative Phrase Mining

In order to effectively extract present keyphrases, we categorize POS tags to the corresponding tag set as the following:

- $\mathcal{T}_{indep}$: {"CD", "FW", "GW", "NN⋆", "VB⋆", "JJ⋆", "RB⋆", "ADD"}

- $\mathcal{T}_{dep}$: {"CC", "POS", "HYPH", "IN"}

- $\mathcal{T}_{start\_dep}$: {"RP"}

- $\mathcal{T}_{end\_dep}$: {"DT", "AFX", "LS"}

where each tag is defined in the NLTK library. An asterisk (*) refers to any character(s) that come after the tag name. For example, "JJ", "JJR", and "JJS" are adjectives, comparative adjectives, and superlative adjectives, respectively, and all of them are elements of the independent tag set $\mathcal{T}_{indep}$.

## A.2  Hyperparameter Search

We perform a grid search to find the best hyperparameter configuration and report the tuning range used for our experiments in Table 7. The evaluation on the validation set is performed for every 5,000 gradient accumulating steps, and the tolerance increases by 1 when the validation loss or $F_1@M$ is worse than the previous evaluation.

| Model | Hyperparameter | Range | Best |
|---|---|---|---|
| BART$_{BASE}$ | learning rate | { 5e-5, 1e-4 } | 5e-5 |
| | warm-up ratio | { 0.0, 0.1 } | 0.0 |
| | batch size | { 4, 8, 16 } | 8 |
| | $\tau$ | { 0.1, 0.3, 0.5, 0.7, 1.0 } | 0.1 |
| | $\lambda$ | { 0.1, 0.2, ..., 1.0 } | 0.3 |
| | epoch | { 10 } | 10 |
| | max tolerance | { 10 } | 10 |
| | max grad norm | { 1.0 } | 1.0 |
| BERT$_{BASE}$ | learning rate | { 1e-5, 2e-5, 3e-5, 4e-5, 5e-5 } | 3e-5 |
| | warm-up ratio | { 0.0, 0.1 } | 0.1 |
| | batch size | { 4, 8, 16 } | 8 |
| | $\tau$ | { 0.1 } | 0.1 |
| | epoch | { 10 } | 10 |
| | max tolerance | { 10 } | 10 |
| | max grad norm | { 1.0 } | 1.0 |

Table 7: Hyperparameter tuning range and best values used in the experiments.