# OpenReview forum: "SimCKP: Simple Contrastive Learning of Keyphrase Representations"
_EMNLP/2023/Conference — EMNLP 2023 Findings_

### Official Review · Reviewer_h8yT · 2023-08-02

**Soundness:** 4

**Excitement:**

4: Strong: This paper deepens the understanding of some phenomenon or lowers the barriers to an existing research direction.

**Missing References:**

[4] Capturing Global Informativeness in Open Domain Keyphrase Extraction - Sun et al. NLPCC 2021

[5] Importance Estimation from Multiple Perspectives for Keyphrase Extraction - Song et al. EMNLP 2021

[6] Learning to extract from multiple perspectives for neural keyphrase extraction - Song et al. Computer Speech & Language 2023

Other missing references:

[2] Addressing Extraction and Generation Separately: Keyphrase Prediction With Pre-Trained Language Models - Liu et al. IEEE 2021 (also combines extraction and generation in a certain manner)

[7] KPDrop: Improving Absent Keyphrase Generation - Ray Chowdhury et al. EMNLP Findings 2022

[8] Learning Rich Representation of Keyphrases from Text - Kulkarni et al. NAACL Findings 2022

[9] Heterogeneous Graph Neural Networks for Keyphrase Generation - Ye et al. EMNLP 2021

[10] Representation Learning for Resource-Constrained Keyphrase Generation - Du et al. EMNLP Findings 2022

**Paper Topic And Main Contributions:**

The authors combine keyphrase extraction (of present keyphrases) and generation (of absent keyphrases) in a unified setting. Unlike prior methods, the authors motivate the use of contrastive learning (rather than sequence labeling) for extraction (using the contextualized encoder representations from BART) combined with the MLE generation loss (for absent keyphrase generation) in a multi-task manner. Furthermore, the authors use a BERT-based re-ranker (trained using contrastive learning again)  to rerank overgenerated absent keyphrases and select the top keyphrases that pass a threshold (tuned as a hyperparameter). The combined method outperforms existing generation models by a high margin.

**Questions For The Authors:**

1. Will the code be released?

2. "F1@5 measures only the top five predictions, but if the model predicts less than five keyphrases, we randomly append incorrect 428
keyphrases until it obtains five" -- this motivation does not make sense to me. Making changes to an evaluation purely for the sake of making it different from another evaluation sounds ill-formed. I can just add some random noise to make it different - but presumably that would be not meaningful. Adding dummy keyphrases would lead to penalizing perfect predictions when ground truth has less than 5 keyphrases. It seems like an ill-motivated modification by all account. (I won't deduct any point from your paper - in particular - because of this - my criticism is aimed more generally against the whole literature following Chan et al. -- still I would be open to hearing if there is a counter to my point here)

**Reasons To Accept:**

1. Reasonably novel and technically sound combination of methods for unified generation/extraction.
2. Reranking after over-generation is a pretty reasonable strategy for improving absent keyphrase generation.

**Reasons To Reject:**

1. Not sure if the sequence labeling baselines are implemented in the best way. The paper says the reason for the poor performance of sequence labeling is "due to the employment of a subword tokenizer, as it requires having to predict more tokens.". However, standard implementations [1] of sequence labeling use some pooling method (for example mean pool, or selecting the first subword token within a word) to transform subwords to words before BIO predictions. It is also standard to use a CRF for sequence labeling. Since one of the main claims in the paper is the superiority of contrastive learning over sequence labeling, the lack of attention to effort in creating stronger sequence labeling baselines is a critical miss. There are also some very strong sequence labeling results in the literature [2, 3].
2. Misses an entire branch of related works that do a sort of contrastive learning [4-6] for keyphrase extraction. This also means that the contrastive learning setup for extraction is not as novel as it may seem.


[1] BERT: Pre-training of Deep Bidirectional Transformers for Language Understanding - Devlin et al. ACL 2019 (even in one of the earliest pre-trained models using subword tokenizer see "We use the representation of the first sub-token as the input to the token-level
classifier over the NER label set". I have seen other papers trying different poolings.

[2] Addressing Extraction and Generation Separately: Keyphrase Prediction With Pre-Trained Language Models - Liu et al. IEEE 2021

[3] SaSAKE: Syntax and Semantics Aware Keyphrase Extraction from Research Papers - Santosh et al. COLING 2020


**Post Rebuttal:** My concerns are mostly addressed.

**Reproducibility:**

3: Could reproduce the results with some difficulty. The settings of parameters are underspecified or subjectively determined; the training/evaluation data are not widely available.

**Reviewer Confidence:**

4: Quite sure. I tried to check the important points carefully. It's unlikely, though conceivable, that I missed something that should affect my ratings.

**Typos Grammar Style And Presentation Improvements:**

A better motivation for overgeneration + reranking should be the recorded high recall performance of beam search [7,8]. Even now the recall is much higher (so there is still room for improvement purely by better reranking).

**Post Rebuttal:** I increased my soundness score to 4 (from 2)  based on the discussion below.

---

> ### Author Rebuttal · Authors · 2023-08-28
>
> We are glad that Reviewer h8yT found our approach “reasonably novel” and “technically sound.” We are deeply thankful for the reviewer’s precious time and critical feedback, which we respond to below:
> - *Sequence Labeling*: We agree that the sequence labeling baseline was naively implemented and that stronger baseline results should be reported to prove the superiority of contrastive learning over sequence labeling. Therefore, we have conducted an additional experiment using BiLSTM-CRF, a widely used strong sequence labeling method. As shown below, the baseline results have been significantly boosted from the naive 3-way classifier; however, they are still outperformed by SimCKP, demonstrating the superiority of our model. We will update this result in our revised manuscript.
> > | Method | In-domain F1@5 | In-domain F1@M | Out-of-domain F1@5 | Out-of-domain F1@M |
> | --- | --- | --- | --- | --- |
> | CL=>3-way classifier | 0.156 | 0.257 | 0.117 | 0.181 |
> | CL=>BiLSTM-CRF | *0.207* | *0.345* | *0.174* | *0.272* |
> | SimCKP | **0.426** | **0.427** | **0.411** | **0.412** |
>
> - *Missing Related Work*: We apologize for missing out on some related work in keyphrase extraction using contrastive learning. We will add the mentioned work, in addition to other studies we might have missed. In terms of novelty, applying contrastive learning for extraction itself is not new; however, all of the aforementioned and previous works have not explored a joint contrastive learning setup for extraction and generation. As we have stated in the introduction, presenting contrastive learning to a unified keyphrase prediction task is one of the main contributions of our work.
> - *Reproducibility*: We recognize that our code would be useful for ongoing research efforts. We plan to release the source code, data, model checkpoints, and prediction results.
> - *Thoughts on F1@5*: We appreciate the reviewer for bringing this to the discussion. We also believe that appending incorrect keyphrases is counterintuitive, as it may represent the model performance as lower than it actually is when the ground truth has less than 5 keyphrases. However, because the literature following Chan et al. measures their models using this procedure, our work follows the same for a fair comparison. With all that said, we do feel that a better evaluation metric is worth exploring for future research.

---

### Official Review · Reviewer_8xSg · 2023-08-03

**Soundness:** 3

**Excitement:**

3: Ambivalent: It has merits (e.g., it reports state-of-the-art results, the idea is nice), but there are key weaknesses (e.g., it describes incremental work), and it can significantly benefit from another round of revision. However, I won't object to accepting it if my co-reviewers champion it.

**Paper Topic And Main Contributions:**

The topic of this paper is about approaches for data- and compute efficiency.
The main contributions of this paper are as follows:
(1)	This paper presents a contrastive learning framework to extract and generate keyphrases by building context-aware phrase-level representations.
(2)	The paper develop a reranker based on the semantic alignment with the document to improve the absent keyphrase prediction performance.
(3)	Experimental results on multiple benchmark datasets demonstrate that the proposed approach is effective.


**Questions For The Authors:**

 (1) The innovation of the paper using the joint learning framework is limited. Existing methods proposed jointly learns to extract and generate keyphrases, such as UniKeyphrase etc.

 (2) Why is contrastive learning framework beneficial for learning phrase-level representations? Please give details.

 (3)  In Table 3, why is the best result of UniKeyphrase (beam=4) not used as the comparison of experimental results? Please give details.

 (4) The paper sets the maximization agreement in both stages. Is it the same to set the maximization agreement measure?

**Reasons To Accept:**

 The proposed method seems to achieve good results on multiple benchmark datasets for keyphrase prediction.

**Reasons To Reject:**

 (1) The innovation of the paper using the joint learning framework is limited. Existing methods proposed jointly learns to extract and generate keyphrases, such as UniKeyphrase etc.

 (2) Why is contrastive learning framework beneficial for learning phrase-level representations? Please give details.

 (3)  In Table 3, why is the best result of UniKeyphrase (beam=4) not used as the comparison of experimental results? Please give details.

 (4) The paper sets the maximization agreement in both stages. Is it the same to set the maximization agreement measure?


**Reproducibility:**

2: Would be hard pressed to reproduce the results. The contribution depends on data that are simply not available outside the author's institution or consortium; not enough details are provided.

**Reviewer Confidence:**

4: Quite sure. I tried to check the important points carefully. It's unlikely, though conceivable, that I missed something that should affect my ratings.

---

> ### Author Rebuttal · Authors · 2023-08-28
>
> We genuinely appreciate Reviewer 8xSg for their thorough and thoughtful review. The reviewer asked several insightful questions, which we respond to below:
> - *Innovation of the Joint Learning Framework*: We certainly agree that a joint learning framework is not new. However, building a joint learning framework is not the main contribution of our work. We have leveraged joint learning because as we stated in the abstract and introduction, the search space in KE is much smaller than in KG, and therefore it is much more effective to combine both approaches for keyphrase prediction. We want to emphasize that our paper is mainly a work that addresses two limitations of the current joint models – sequence labeling and MLE – and presents contrastive learning as a solution.
> - *Benefits of Contrastive Learning*: We appreciate the reviewer for pointing this out. As stated in the paper, our contrastive learning framework brings the document and keyphrase embeddings together in the semantic space while pushing away the rest of the phrase embeddings from the document. Hence, we can say that our model has learned to distinguish representations at a *phrase level*. At inference time, computing the distance (cosine similarity in our case) between document and phrase embeddings will be equivalent to the scoring of the phrases. Such scores will then be ranked, and the top *k* are successfully retrieved. A related work [1] explained this contrastive learning approach as “introducing competition between the candidate keyphrases to extract more salient keyphrases.” Furthermore, learned phrase-level representations via contrastive learning are especially effective in scenarios where phrases overlap. For example, say that only one of the overlapping phrases “boundary integral equations” and “integral equations” is the ground-truth keyphrase. For a standard language model, the two phrases are semantically similar; therefore, the model is very likely to predict both as keyphrases for a given document. However, for our model, contrastive learning has clearly distinguished their representations w.r.t. the document, thereby successfully predicting the one true keyphrase. This motivation may not have been clearly conveyed in the paper; thus, we will update this explanation in our revised manuscript.
> - *UniKeyphrase Results*: The authors of UniKeyphrase [2] and PromptKP [3] are the same, and thus we followed the convention in the PromptKP paper, which compared with the greedy search results. However, we agree that the best results should be used for comparison. We will revise our manuscript accordingly.
> - *Maximization Agreement*: We apologize if we misunderstood the question. The maximization agreement in both stages uses the same contrastive loss in Equation 3. The only difference is that present keyphrase representations are utilized in Stage 1, whereas the absent keyphrases are used in Stage 2.
>
> [1] Importance Estimation from Multiple Perspectives for Keyphrase Extraction (Song et al., 2021)
>
> [2] UniKeyphrase: A Unified Extraction and Generation Framework for Keyphrase Prediction (Wu et al., 2021)
>
> [3] Fast and Constrained Absent Keyphrase Generation by Prompt-Based Learning (Wu et al., 2022)

---

### Official Review · Reviewer_TNRt · 2023-08-03

**Soundness:** 2

**Excitement:**

3: Ambivalent: It has merits (e.g., it reports state-of-the-art results, the idea is nice), but there are key weaknesses (e.g., it describes incremental work), and it can significantly benefit from another round of revision. However, I won't object to accepting it if my co-reviewers champion it.

**Paper Topic And Main Contributions:**

The paper proposes to generate the keyphrase for a given document using a simple contrastive learning framework.

**Reasons To Accept:**

1. The author applies a simple contrastive learning framework to generate keyphrases for a given document, which makes total sense.
2. The experimental setup is described and the experiments show the proposed method seems to achieve good results on five datasets.
3. The paper is well written.

**Reasons To Reject:**

1. The motivation of the work is very weak. Why the contrastive learning framework would benefit learning phrase-level representations (or benefit observing and scoring keyphrases as a whole)?

2. The source code for training the proposed model and the predicted results are not released. It would be useful for ongoing research efforts if the paper could release its source code and predicted results.

3. In Table 3, the best results of UniKeyphrase (beam=4) should be used for comparing different models.


**Reproducibility:**

2: Would be hard pressed to reproduce the results. The contribution depends on data that are simply not available outside the author's institution or consortium; not enough details are provided.

**Reviewer Confidence:**

5: Positive that my evaluation is correct. I read the paper very carefully and I am very familiar with related work.

---

> ### Author Rebuttal · Authors · 2023-08-28
>
> We are encouraged that Reviewer TNRt found our paper making “total sense” and “well written.” We deeply appreciate the reviewer’s positive comments and helpful feedback, which we respond to below:
> - *Motivation*: We appreciate the reviewer for pointing this out. As stated in the paper, our contrastive learning framework brings the document and keyphrase embeddings together in the semantic space while pushing away the rest of the phrase embeddings from the document. Hence, we can say that our model has learned to distinguish representations at a *phrase level*. At inference time, computing the distance (cosine similarity in our case) between document and phrase embeddings will be equivalent to the scoring of the phrases. Such scores will then be ranked, and the top *k* are successfully retrieved. A related work [1] explained this contrastive learning approach as “introducing competition between the candidate keyphrases to extract more salient keyphrases.” Furthermore, learned phrase-level representations via contrastive learning are especially effective in scenarios where phrases overlap. For example, say that only one of the overlapping phrases “boundary integral equations” and “integral equations” is the ground-truth keyphrase. For a standard language model, the two phrases are semantically similar; therefore, the model is very likely to predict both as keyphrases for a given document. However, for our model, contrastive learning has clearly distinguished their representations w.r.t. the document, thereby successfully predicting the one true keyphrase. This motivation may not have been clearly conveyed in the paper; thus, we will update this explanation in our revised manuscript.
> - *Reproducibility*: We recognize that our code and predicted results would be useful for ongoing research efforts. We plan to release the source code, data, model checkpoints, and prediction results.
> - *UniKeyphrase Results*: The authors of UniKeyphrase [2] and PromptKP [3] are the same, and thus we followed the convention in the PromptKP paper, which compared with the greedy search results. However, we agree that the best results should be used for comparison. We will revise our manuscript accordingly.
>
> [1] Importance Estimation from Multiple Perspectives for Keyphrase Extraction (Song et al., 2021)
>
> [2] UniKeyphrase: A Unified Extraction and Generation Framework for Keyphrase Prediction (Wu et al., 2021)
>
> [3] Fast and Constrained Absent Keyphrase Generation by Prompt-Based Learning (Wu et al., 2022)

---

### Official Review · Reviewer_zjaV · 2023-08-05

**Soundness:** 3

**Excitement:**

3: Ambivalent: It has merits (e.g., it reports state-of-the-art results, the idea is nice), but there are key weaknesses (e.g., it describes incremental work), and it can significantly benefit from another round of revision. However, I won't object to accepting it if my co-reviewers champion it.

**Paper Topic And Main Contributions:**

The paper introduces an interesting approach towards the tasks of keyphrase extraction-generation. The framework designed by the author comprises of a multi-task model and a reranker model. The evaluation results presented are compared with standard keyphrase evaluation state of the art papers across popular datasets. Appropriate ablation study has been performed after the results.

**Questions For The Authors:**

Question A: do you plan to make the code, models, data from the negative phrase mining, available to the general public?

**Reasons To Accept:**

> New framework with multi-task learning objective and reranking to boost evaluation metrics.

>Achieves new state of the art performance over popular keyphrase extraction/generation datasets.

> Ablation studies have been performed to strengthen authors' argument towards novel contribution.

**Reasons To Reject:**

> New framework seems computationally very expensive. It aggregates/boosts results from separate individual models. Comparing it to standard keyphrase generation papers in table 2 and 3 is not exactly apples to apples comparison, given they are standalone RNN or transformer based models.

> The negative phrase mining, new learning objective and reranker are a combination of specific heuristics proposed by the authors. If the code, data and models are not made available to the public, it may cause serious reproducibility issues.

**Reproducibility:**

2: Would be hard pressed to reproduce the results. The contribution depends on data that are simply not available outside the author's institution or consortium; not enough details are provided.

**Reviewer Confidence:**

5: Positive that my evaluation is correct. I read the paper very carefully and I am very familiar with related work.

---

> ### Author Rebuttal · Authors · 2023-08-28
>
> We sincerely appreciate Reviewer zjaV for their valuable time and constructive feedback on improving our paper. The reviewer raised several concerns, which we respond to below:
> - *Computational Cost*: As the reviewer pointed out, the training cost of SimCKP is approximately twice as much as other standalone models, as it does require additional training of the reranker (e.g., BERT). However, compared to the model sizes and GPU power available today, we believe that the cost is not burdensome. Furthermore, the inference cost includes retrieving top *k* phrases in addition to beam search, which is not extremely expensive. Therefore, deploying it to real-world applications by any chance will not be a big problem. In regards to an apples-to-apples comparison with other models, the salient sentence selector in SEG-Net [1] and the stacked relation layer in UniKeyphrase [2] are also additional modules that require extra training. In that sense, we believe that our model is not much different from theirs. Still, to make the comparison as fair as possible, we have incorporated base models for our framework.
> - *Reproducibility*: We recognize that data from the negative phrase mining, as well as the code and model checkpoints, are crucial for reproducing the paper. We plan to release the source code, data, model checkpoints, and prediction results.
>
> [1] Select, Extract and Generate: Neural Keyphrase Generation with Layer-wise Coverage Attention (Ahmad et al., 2021)
>
> [2] UniKeyphrase: A Unified Extraction and Generation Framework for Keyphrase Prediction (Wu et al., 2021)

---

### Meta-Review · Area_Chair_4T7L · 2023-09-21

**Recommendation:** 3

**Metareview:**

This paper presents a joint learning framework based on contrastive learning to extract and generate keyphrases. Concerns lie in the novelty over UniKeyPhrase and the motivation of the contrastive learning framework (partially addressed in the rebuttal). Comparison under the beam search setting is also needed.

---

### Decision · Program_Chairs · 2023-10-07

**Decision:**

Accept-Findings

**Comment:**

This paper presents a joint learning framework based on contrastive learning to extract and generate keyphrases. Concerns lie in the novelty over UniKeyPhrase and the motivation of the contrastive learning framework (partially addressed in the rebuttal). Comparison under the beam search setting is also needed.